# Sedation Therapy in Intensive Care Units: Harnessing the Power of Antioxidants to Combat Oxidative Stress

**DOI:** 10.3390/biomedicines11082129

**Published:** 2023-07-28

**Authors:** Gen Inoue, Yuhei Ohtaki, Kazue Satoh, Yuki Odanaka, Akihito Katoh, Keisuke Suzuki, Yoshitake Tomita, Manabu Eiraku, Kazuki Kikuchi, Kouhei Harano, Masaharu Yagi, Naoki Uchida, Kenji Dohi

**Affiliations:** 1Department of Emergency, Disaster and Critical Care Medicine, School of Medicine, Showa University, 1-5-8 Hatanodai, Shinagawa-ku, Tokyo 142-8555, Japan; gen.musashi83@gmail.com (G.I.);; 2Department of Emergency Medicine, School of Medicine, The Jikei University, 3-25-8 Nishishinbashi, Minato-ku, Tokyo 105-8461, Japan; 3Center for Instrumental Analysis, School of Pharmacy, Showa University, 1-5-8 Hatanodai, Shinagawa-ku, Tokyo 142-8555, Japan; 4Clinical Research Institute for Clinical Pharmacology and Therapeutics, Showa University Karasuyama Hospital, 6-11-11 Kitakarasuyama, Setagaya-ku, Tokyo 157-8577, Japan

**Keywords:** oxidative stresses, neuroinflammation, traumatic brain injuries, intensive care, sedatives, antioxidant effects, biologic monitoring, barbiturates, electron spin resonance, translational research

## Abstract

In critically ill patients requiring intensive care, increased oxidative stress plays an important role in pathogenesis. Sedatives are widely used for sedation in many of these patients. Some sedatives are known antioxidants. However, no studies have evaluated the direct scavenging activity of various sedative agents on different free radicals. This study aimed to determine whether common sedatives (propofol, thiopental, and dexmedetomidine (DEX)) have direct free radical scavenging activity against various free radicals using in vitro electron spin resonance. Superoxide, hydroxyl radical, singlet oxygen, and nitric oxide (NO) direct scavenging activities were measured. All sedatives scavenged different types of free radicals. DEX, a new sedative, also scavenged hydroxyl radicals. Thiopental scavenged all types of free radicals, including NO, whereas propofol did not scavenge superoxide radicals. In this retrospective analysis, we observed changes in oxidative antioxidant markers following the administration of thiopental in patients with severe head trauma. We identified the direct radical-scavenging activity of various sedatives used in clinical settings. Furthermore, we reported a representative case of traumatic brain injury wherein thiopental administration dramatically affected oxidative-stress-related biomarkers. This study suggests that, in the future, sedatives containing thiopental may be redeveloped as an antioxidant therapy through further clinical research.

## 1. Introduction

The extent to which the oxidant–antioxidant balance in the body is disrupted by the generation of excess reactive oxygen species (ROS)/reactive nitrogen species (RNS) is called oxidative stress [1,2]. Oxidative stress plays an important role in the pathogenesis of many acute diseases, including neurological emergencies as well as age-related diseases, such as arthritis, diabetes, dementia, cancer, atherosclerosis, vascular diseases, obesity, osteoporosis, and metabolic syndromes [3,4,5]. ROS/RNS are produced in vivo via oxygen and regulatory cellular activities like cell survival, stress response, and inflammation [5]. Among ROS, superoxides (O_2_^•−^), hydroxyl radicals, and nitric oxide (NO) are free radicals with unpaired electrons. Because of their strong biological reactivity, they are major participants in inflammatory reactions. NO production in vivo is mediated by NO synthase (NOS). Among the three NOS isozymes, overproduced NO via the expression of inducible NOS (iNOS) has a role as an inflammation mediator [6,7]. The reactivity of ONOO^−^, which is generated via the reaction of O_2_^•−^ and NO, is also crucial [8,9]. Oxidative stress also plays an important role during various acute crisis situations, including acute inflammation, surgical stress, ischemia-reperfusion, and trauma [10]. Recently, oxidative stress was reported to play a role in COVID-19 infection [11]. A case series of COVID-19 patients treated with vitamin C reported decreased mortality, significant reductions in inflammatory markers, and a trend toward decreased oxygen demand [12]. Obvious methods to control the critical situation caused by this imbalance are to balance or eliminate the generation of ROS [10]. Numerous animal studies have demonstrated that the administration of free radical scavengers and antioxidants dramatically reduces organ damage [10,13]. Although experimental studies showed positive results, there is very little evidence that antioxidant therapy is clinically beneficial, and few antioxidants have been used in clinical settings. Oxygen consumption in the brain is very high, and neurological emergencies are considered typical conditions in which antioxidant therapy may be useful. Edaravone, a hydroxyl radical scavenger, is the only clinically applicable agent for ischemic stroke [14,15,16,17,18]. Therefore, there is a critical need to develop clinically available antioxidants and new antioxidant therapies.

Sedatives are widely used in acutely critically ill patients, generally to maintain sedation in intensive care units. Sedation therapy in ICUs is performed for sedation and to decrease oxygen consumption and metabolism throughout the body [19]. Some sedatives are known to have beneficial pharmacological effects in addition to simple anesthetic effects [20]. Moreover, sedatives are also used during neurological emergencies for patients with increased intracranial pressure and status epilepticus. In patients with head trauma, barbiturates decrease cerebral oxygen consumption and cerebral blood flow and correspondingly decrease intracranial pressure (ICP). In addition to decreased ICP, increased partial pressure of oxygen in brain tissue (PbtO_2_) and decreased excitatory amino acids have been reported [21]. Propofol (2,6-diisopropylphenol) is reported to have powerful antioxidant properties, having a chemical structure similar to that of the endogenous antioxidant α-tocopherol (vitamin E) [20].

Free radicals are products of normal cellular metabolism [1,2,3,4,5]. When cells utilize oxygen, redox processes produce free radicals, usually ROS and RNS [22]. Free radicals can be defined as molecular bodies or molecular fragments that can exist independently. Free radicals have one or more unpaired electrons in their outer atomic or molecular orbitals [22]. They are described as “free radical” and occur in different types, such as superoxides (O_2_^•−^), hydroxyl radicals (OH^•^), alkoxyl radicals (RO^•^), peroxyl radicals (ROO^•^), nitric oxide (nitrogen monoxide) (NO^•^), and nitrogen dioxide (NO_2_^•^) [23]. The biological effects of each type are known to differ [24], and the free radicals that can be scavenged or inhibited by each antioxidant are different. For example, edaravone, a free radical scavenger used clinically for cerebral infarction, scavenges hydroxyl radicals and nitric oxide (NO) but not superoxide radicals [14]. Therefore, it is crucial to investigate the scavenging ability of antioxidants for each type of free radical. To the best of our knowledge, to date, no study has evaluated the scavenging ability of antioxidant sedatives, such as propofol, for different types of free radicals.

The purpose of this study was to investigate the potential future clinical applications of sedatives as novel antioxidant therapies. Specifically, we used an in vitro electron spin resonance (ESR) assay to investigate whether common clinical sedatives have direct free-radical-scavenging activity for different types of free radicals. Moreover, we presented clinical cases of patients with head trauma treated with thiopental-based barbiturates and described the changes in in vivo oxidative antioxidant biomarkers.

## 2. Materials and Methods

### 2.1. Reagents

Xanthine oxidase (XOD), hypoxanthine (HPX), and diethylene triamine penta-acetic acid (DETAPAC) were obtained from Sigma Chemical (St. Louis, MO, USA). The spin trap 5,5-dimethyl-1-pyrroline-*N*-oxide (DMPO) was obtained from Labotec (Tokyo, Japan). 1-Hydroxy-2-oxo-3-(*N*-3-methyl-3-aminopropyl)-3-methyl-1-triazene (NOC-7), 2-(4-carboxyphenyl)-4,4,5,5-tetramethylimidazoline-1-oxyl-3-oxide, and sodium salt (carboxy-PTIO) were obtained from Dojin Chemical (Kumamoto, Japan). A superoxide dismutase (SOD) standard solution kit was purchased from Labotec. Sumatriptan succinate (GR43175 C) was gifted from Glaxo Wellcome (London, UK).

### 2.2. In Vitro ESR Method

The ESR analysis of the spin adduct was performed at room temperature using a JES-REIX X-band spectrometer (JEOL, Tokyo, Japan). The following ESR measurement conditions were implemented: magnetic field of 335.6 ± 5.0 mT; microwave power of 8.0 mW; sweep time of 2 min/0.03 s; and modulation amplitude of 0.1 mT. Manganese oxide was used as an external standard because it provided a constant signal against which all peak heights were compared.

To calculate the relative peak height, the sample peak height was divided by the manganese oxide peak height.

The superoxide radical (O_2_^•−^) was generated with a hypoxanthine XOD system. For the assay, 50 µL of 2 mM hypoxanthine in phosphate-buffered saline, 20 µL of 0.5 mM DETAPAC, 30 µL of each sedative dissolved in dimethyl sulfoxide, 50 µL of 0.46 M DMPO, and 30 µL of 0.5 U/mL XOD were mixed in a test tube. The solution was placed in a special flat cell in which DMPO–superoxide, the spin adduct, was analyzed using ESR (Figure 1a) [24,25].

A standard curve was constructed with 0.4 to 40 U/mL of superoxide dismutase (SOD) added to the system instead of each sedative.

Hydroxyl radical (OH^•^)-scavenging activity was also measured using the ESR spin trap method (Figure 1b). The reaction mixture comprised 50 µL of 92 mM DMPO, 50 µL of 1 mM FeSO_4_, 0.02 mM DETAPAC, 50 µL of 1 mM hydrogen peroxide, and 50 µL of each sedative. After rapid stirring, the reaction mixture was placed into an ESR flat cell. Recording of the ESR spectrum was started 60 s after the addition of 1 mM H_2_O_2_ [24].

NO-scavenging activity was estimated using carboxy-PTIO. NO was generated from NOC-7. All reagents were dissolved in a 0.1 M phosphate buffer (pH 7.4) except NOC-7, which was diluted to 0.1 mM in a 0.1 N NaOH solution. First, 20 µL of each sedative was added to 140 µL of a 0.1 M potassium buffer followed by 20 µL of 0.1 mM carboxy-PTIO and 20 µL of 0.1 mM NOC-7. Immediately after vortex mixing, the sample solution was transferred into a flat cell (200 µL capacity). ESR measurements were started 1, 3, 5, 7, 10, 12, and 15 min after the addition of NOC-7 (Figure 2) [14].

Singlet oxygen was generated via photosensitization reactions with rose bengal. Singlet oxygen was indirectly estimated as the peak intensity of the 2,2,6,6-tetramethyl-4-4-hydroxy-piperidinyloxy (4-OH TEMPO) radical produced via the oxidation of 2,2,6,6-tetramethyl-4-hydroxy-piperidine (4-OH TEMP) with singlet oxygen (produced via photosensitization with rose bengal) using ESR. Samples were diluted in dimethyl sulfoxide (DMSO) to the required final concentrations. The samples (in 60 µL of DMSO), 10 μL of 1 mM DTPA, 0.1 M phosphate buffer (pH 7.4; 70 µL), 100 mM 4-OH TEMP (40 µL), and 200 µM rose bengal (20 µL) were irradiated for 3 min (1.57 J/cm^2^) with a green-light-emitting diode with a λMAX of 540 nm (Simantec Ltd., Tokyo, Japan) [26].

### 2.3. Statistical Analysis

Data were analyzed using JMP 17 statistical software. Data are presented as means ± standard error. All experiments were performed in triplicate, except for some tests with little or no intensity change. A one-way analysis of variance test was performed to compare. Significance was defined as *p*  <  0.05 *.

## 3. Results

The effects of Propofol and Thiopental on ESR signals of and superoxide radical (O_2_^•−^) and hydroxyl radical (OH^•^) are shown in Figure 3 and Figure 4. Regarding superoxide radical -scavenging activity, ESR showed that the formation of the superoxide radical–DMSO spin adduct was strongly inhibited with 2.06 mM and 20.63 mM thiopental. DEX and propofol did not have direct superoxide-radical-scavenging activity (Figure 3, Figure 5a and Figure 6a).

Regarding hydroxyl radical-scavenging activity, ESR showed that the formation of the hydroxyl radical–DMSO spin adduct was inhibited with 0.11 mM DEX strongly inhibited with 1.40 mM and 14.02 mM propofol and with 2.58 mM and 25.79 mM thiopental (Figure 4, Figure 5b and Figure 6b).

Regarding nitric oxide (NO)-scavenging activity, the penta-signal of carboxy-PTIO disappeared and the hepta-signal of carboxy-PTI appeared in the ESR recording after the addition of NOC-7, a NO generator. The appearance of the hepta-signal of carboxy-PTI was inhibited with 11.22 mM propofol and was strongly inhibited with 2.06 mM and 20.63 mM thiopental (Figure 5c).

Regarding singlet oxygen (^1^O_2_)-scavenging activity, propofol and thiopental scavenged singlet oxygen, 10.32 mM thiopental completely inhibited singlet oxygen production, and DEX did not have singlet oxygen-scavenging activity (Figure 5d and Figure 6c).

In summary, propofol, a previously known antioxidant, had direct scavenging activity for hydroxyl radicals, NO, and singlet oxygen. Furthermore, thiopental, a classical sedative, acted as a strong scavenger of all the free radicals examined, including superoxide radicals. Finally, we demonstrated the previously unknown direct hydroxyl-radical-scavenging activity of DEX.

## 4. Clinical Case

We present a representative case of traumatic brain injury in which thiopental administration dramatically affected oxidative-stress-related biomarkers. A 15-year-old woman was transferred to the emergency department for the trauma inflicted by attempting suicide by jumping onto a train track. Her initial consciousness level was E1V1M4 on the Glasgow Coma Scale (GCS). Dilated pupils were observed, but bilateral contralateral reflections remained slight. Based on these clinical findings, the patient was diagnosed to be in a state of impending cerebral herniation due to severe traumatic brain injury (TBI). Initial brain computed tomography (CT) showed severe diffuse cerebral swelling, acute subdural hemorrhage, and traumatic subarachnoid hemorrhage (Figure 6a). Targeted temperature management therapy to maintain 35 °C and ICP monitoring were immediately started to treat the impending brain herniation associated with increased ICP. At the time of monitor insertion, the ICP exceeded 50 mmHg. After rewarming, the ICP was occasionally elevated above 50 mmHg despite continued normothermia (36 °C). Subsequent CT scans revealed the progressive deterioration of cerebral edema. Consequently, barbiturate coma therapy using thiopental (1 mg/kg/h) was added. Under thiopental administration, ICP was stabilized below 20 mmHg, and the subsequent CT scan on day 10 demonstrated a reduction in cerebral edema. Her consciousness level was improved to E4V5M6/GCS and she was transferred to a psychiatric facility without neurological deficit. With the results of these in vitro ESR experiments, we retrospectively confirmed the in vivo oxidative biomarkers (uric acid (UA), bilirubin (Bil), and carbon-monoxide-binding hemoglobin (COHb)) in this patient. The levels of UA, which is an endogenous antioxidant, were very low before thiopental administration but increased after administration. Moreover, Bil and COHb, metabolites of the stress protein heme oxygenase-1 (HO-1), were decreased after thiopental administration.

## 5. Discussion

Oxidative stress plays an important role in the pathogenesis of many critical illnesses that require intensive care, such as trauma, severe infections, stroke, and ischemic heart disease [3,4,5]. For the proper diagnosis and treatment of these diseases and considering their pathologies, it is important to monitor various redox components during disease progression and to regulate the redox balance of the patient with antioxidants. In this study, we evaluated the direct free-radical-scavenging activity of propofol, thiopental, and dexmedetomidine. The present results indicated that the three sedatives had different patterns of free-radical-scavenging activity. Surprisingly, thiopental, a representative sedative, had direct scavenging activity for all four species investigated: superoxide radicals, hydroxyl radicals, NO, and singlet oxygen.

Various drugs and treatments have been developed to maintain the oxidant–antioxidant balance in the body to reduce the damage caused by oxidative stress [2,15,27,28]. Representative drugs and agents include molecular hydrogen (H_2_), vitamin C, vitamin E, superoxide dismutase (SOD), and edaravone [29,30,31,32]. Meanwhile, with regard to NO, NO donors used as vasodilators and sildenafil used as a treatment for impotence are well known [33,34,35]. A direct NO scavenger, carboxy-2-phenyl-4,4,5,5,-tetramethylimidazoline-1-oxyl 3-oxide (cyboxy-PTIO), has been applied in various studies but has not yet been applied clinically [36]. The pharmacological effects of vitamin C that could make it a potential option for the prevention and treatment of COVID-19 have recently been reviewed. Clinicians using intravenous vitamin C in severely ill COVID-19 patients have reported positive clinical effects upon administration of 3 g every 6 h, together with steroids and anti-coagulants [11,29,37]. One recent topic is hydrogen’s powerful antioxidant properties demonstrated by selectively inhibiting the oxidative effects of the most harmful ROS/RNS, OH and ONOO. H_2_ also has a cellular defense function against oxidative stress, eliminating harmful ROS/RNS in the body and suppressing inflammatory responses. Regardless of the method of administration, such as inhalation, intravenous, or oral, H_2_ has been reported to be effective in a variety of diseases and conditions due to its high rate of transfer to brain tissue [28].

These drugs are used for the primary purpose of scavenging ROS. Meanwhile, some drugs are known to have antioxidant effects as a secondary action. These drugs are used primarily for their other pharmacological effects but actually possess antioxidant and ROS-scavenging properties. Indomethacin (IND) is a strong cyclooxygenase (COX) inhibitor and has been widely used as a nonsteroidal anti-inflammatory drug (NSAID). Recently, COX inhibitors including IND have been shown to have not only antipyretic and anti-inflammatory effects but also various pharmacological effects (table) including protection against neuronal cell death [24,38]. IND also has direct strong free-radical-scavenging activity [24]. Unlike newly developed drugs, these routinely used drugs are already well experienced in clinical use and have advantages in terms of safety and the occurrence of side effects. Furthermore, the prices of the drugs are also lower, making them more economical than developing a new drug. In addition to these NSAIDs, various sedatives are also routinely used in ICUs and in surgery.

Various sedatives are used for intensive care and surgery. Some sedatives are known to have pharmacological effects other than sedation, with several reported to have antioxidant properties [13]. Propofol is known to have antioxidant properties [13]. The molecular structure of propofol is similar to that of α-tocopherol, one of the strongest endogenous antioxidants [20]. Previous studies have reported that propofol inhibits oxidative stress in preclinical and clinical studies [39].

The present study investigated the direct ROS/RNS-scavenging ability of typical sedatives used in clinical practice, including dexmedetomidine, an α2-adrenoceptor agonist drug for which the antioxidant activity has not yet been investigated using ESR methods. It is also very important to clarify the pharmacological mechanism of action of each drug’s antioxidant activity. However, the in vitro ESR method employed in this study can only measure the direct scavenging activity of each ROS and NO radical. The mechanism of the scavenging activity of each drug against each free radical observed in this study requires further investigation.

The results of this study show that propofol scavenges hydroxyl radicals, NO radicals, and singlet oxygen, but not superoxide radicals. Previous in vitro ESR studies have confirmed that propofol inhibits hydroxyl radical generation but not superoxide generation. The present study supports these data and demonstrates the previously unknown ability of propofol to scavenge NO radicals and singlet oxygen.

Dexmedetomidine, a selective and potent alpha 2-adrenergic receptor agonist, was approved by the US Food and Drug Administration for sedation in 1999. In animal studies, dexmedetomidine exerts neuroprotective effects in forebrain ischemia, focal cerebral ischemia, and incomplete forebrain ischemia [40,41]. In this study, DEX had direct scavenging activity for hydroxyl radicals. In recent studies, DEX was found to decrease cerebral ischemia and SCI-induced intracellular ROS production and apoptosis in the brains of rats [42,43]. Akpınar et al. reported that DEX treatment reduces cerebral-ischemia-induced oxidative stress, cell death, and intracellular Ca^2+^ signaling through the inhibition of TRPM2 and TRPV1 [40].

Indeed, barbiturates have been recommended to treat high and refractory ICP since the early 1980s [44,45]. They are still suggested as a second or third line of treatment in US guidelines [46]. Thiopental is a classic sedative with a wide range of uses in serious conditions but is currently less favored in clinical practices. Few studies have examined thiopental and oxidative stress. Barbiturate anesthesia has been reported to inhibit the fatty acid peroxidation of neural tissue after cerebral ischemia and to enhance antioxidant capacity [47,48]. Lee et al. also compared plasma oxidative stress after surgery in dogs sedated with thiopental and propofol [49] and reported lower oxidative stress with propofol than thiopental. The direct ROS/RNS-scavenging potential of thiopental in the present study was found to be dramatic. Thiopental strongly inhibited all the radical species examined in this study. Thiopental scavenged superoxides, hydroxyl radicals, NO, and singlet oxygen in a dose-dependent manner. In addition to its purpose as a sedative, thiopental is sometimes utilized in neurointensive care in patients with decreased ICP and status epilepticus. Thiopental has some serious side effects; for example, pneumonia and hypotension are frequent side effects of barbiturates. The early use of barbiturates was significantly associated with increased ICU mortality [50]. There are few trials that have evaluated barbiturates in severe TBI patients, and none are recent. However, a novel method for the administration of thiopental has been developed to prevent these complications [51,52].

Our data suggest that thiopental has potent superoxide-scavenging activity, which may result in clinically compromised immunity. Within the body, superoxides are a strong bactericidal agent. Additionally, it has been proposed that the thiopental-mediated inhibition of NF-κB induces apoptosis in granulocytes in response to TNF-α stimulation [53]. NF-κB is activated in many cells by a variety of stimulants with redox-regulatory properties, and ROS are involved in activating the NF-κB pathway. ROS were proposed to be involved in the activation of the NF-κB pathway. Clinical reports indicate that following the induction of barbiturate coma for refractory intracranial hypertension, a decrease in white blood cell count is common, occurring in 81% of patients [54]. However, regardless of these complications, superoxide scavenging is an extremely important therapeutic target in neurologic emergencies. Superoxides are considered to be one of the root causes of the production of all types of ROS, oxidative stress activity, and secondary brain damage [27]. Essentially, the cascade of all ROS and lipid peroxidation begins with the generation of superoxide in vivo. Edaravone, which is effective against cerebral infarction, has been applied clinically as a free radical scavenger. However, edaravone only has hydroxyl-radical-scavenging activity, and it does not scavenge superoxides [14,15]. Based on this functionality, further clinical and basic studies on the ROS-scavenging ability of thiopental against various reactive oxygen species, including superoxides, are expected, given that the drug is already used in clinical practice.

Our study reported a case of severe TBI treated with thiopental. The patient received temperature control therapy for an imminent brain herniation due to post-traumatic brain swelling. Initially, the patient was sedated with midazolam. However, because ICP control was very difficult, barbiturate therapy with thiopental was administered. Although the potent ROS-scavenging ability of thiopental was confirmed in this study, endogenous oxidative stress markers such as COHb, Bil, and UA, which were measured with usual blood sampling during the course of this study, were confirmed retrospectively (Figure 7 and Figure 8). Radical-scavenging molecules in vivo include water-soluble ascorbic acid and fat-soluble tocopherols ingested from the diet, and ubiquinone, GSH, Bil, and UA synthesized in vivo [55,56,57,58]. UA, a cause of gout, is also known to function as a radical scavenger and endogenous antioxidant [57]. Bil is also known to be an endogenous antioxidant [56]. Bil production is mediated by the induction of HO-1. HO is induced by various stresses and is recognized as a stress-sensitive marker protein [55,56,57,58]. Bil elevation was observed after a hemorrhagic stroke, reflecting the intensity of the oxidative stress [56]. Higher Bil was an independent protective factor for arteriosclerotic cardiovascular disease and negatively associated with the prognosis of stroke, acute myocardial infarction, and peripheral arterial disease, but positively associated with in-hospital cardiovascular death and major adverse cardiac events [59]. Plasma Bil concentrations serve as a useful marker of oxidative stress in patients with severe neuronal conditions. Biliverdin, CO, and iron are produced from heme by HO-1. Biliverdin is reduced to bilirubin by reductase. Bil and CO produced via these pathways are also known to possess strong antioxidant properties in vivo. In the present case, ICP was reduced, and intracranial hypertension was improved after the administration of thiopental, which has a strong ROS-scavenging capacity. Bil and COHb concentrations followed a similar time course. Both were decreased with thiopental administration. Thiopental may suppress the stress protein HO-1. The trend in the blood concentration of UA, an antioxidant, was markedly increased by the administration of thiopental. It was suggested that the rapid changes in UA concentration may have been influenced by the administration of thiopental, a potent free radical scavenger. It is known that the oxidative stress level increases in the rewarming phase after targeted temperature management therapy [2]. In this case, the results suggest that thiopental may have had some effect on the oxidant–antioxidant balance in the body. At present, thiopental is used to control intracranial hypertension in neurological emergencies [60]. A patient had markedly elevated ICP; hence, thiopental was administered to prevent cerebral herniation. Thiopental was selected because it can decrease cerebral blood flow, which was observed in CT images showing diffuse cerebral swelling owing to increased cerebral blood flow. Until this observation was noted, its antioxidant effect was not expected. The patient’s UA, an endogenous antioxidant, was initially very low, which might have been due to the induction of excessive oxidative stress. In such cases, the administration of thiopental, which has antioxidant properties, may be a better choice than other sedative agents. Alternate clinically relevant methods to assess oxidative stress are required to choose a relevant therapy for a patient.

This study was only a preliminary case presentation. In clinical practice, data may be influenced by a variety of factors and the condition of the patient prevailing at the time. Therefore, although the changes in UA, CO, and bilirubin data may not be solely due to thiopental administration, no other oxidative-stress-related treatment was being administered to this patient at the time the data were collected. Moreover, there were no pathological changes that would cause such a drastic change.

A major reason for the elementary clinical research on oxidative stress and the development of treatments is that it is very difficult to monitor oxidative stress and the oxidant–antioxidant balance [61]. Basically, the in vivo monitoring of free radicals is extremely difficult to apprehend due to their reaction times. Therefore, most previous studies have discussed clinical results and indirect biomarkers. In a previous study, we used ex vivo ESR to measure alkoxyl radicals in the blood and successfully developed a monitoring method [15]. However, this method could not be implemented at all facilities as the method was complex. In another study, the d-ROMs (diacron reactive oxygen metabolites) method was used to measure hydroperoxide, one of the reactive oxygen metabolites in the blood, to determine the antioxidant effect of hypothermia [2]. Although this monitoring method was simple and practical, it was difficult to determine the origin of these metabolites produced by organs, making it difficult to examine the overall oxidant–antioxidant balance. Tanaka et al. reviewed the use of redox biomarkers in multiple sclerosis (MS), a demyelinating disease of the central nervous system [62]. Reactive chemical species, oxidative enzymes, antioxidants, antioxidant enzymes, degradation products, and end products are potential biomarkers of MS, which can allow early detection and secondary prevention, as well as suggest a possible clinical course, predict MS patients’ responses to specific treatments, and provide treatment targets. However, it is very difficult to accurately measure numerous oxidative antioxidant biomarkers, each with different clinical implications, in all patients. The sedative agents evaluated in this study are typical drugs routinely used in ICUs and other settings. These drugs, on top of their well-known pharmacological effects, can potentially influence ROS/RNS, a key factor for many acute diseases. Sedatives may affect a patient’s oxidant–antioxidant balance, as observed in the present case. Free-radical-scavenging capacity may have a negative impact on a patient’s condition, just as the suppression of superoxide radicals may affect the immune system of a patient with an infectious disease. Therefore, clinicians need to understand the role of oxidative stress in each patient’s pathophysiology and the benefits of controlling the oxidant–antioxidant balance when selecting sedatives. To develop novel antioxidant therapies and drugs in the future, a monitoring system is required to solve these issues. Further studies are needed to explore the potential of sedation therapy as an antioxidant as well as its potential use for the control of ICP.

## 6. Conclusions and Future Perspectives

In this study, we reported that various commonly used sedatives scavenge different collections of radicals. In this study, we found that dexmedetomidine has direct hydroxyl-radical-scavenging activity and thiopental has very potent ROS/RNS-scavenging activity. These results suggest the possibility that this new antioxidant therapy can be developed for clinical application in the future. Each radical plays a different role in various pathologies. Physicians involved in critical care may need to understand the pharmacological properties of each sedative agent as a potential free radical scavenger with respect to other considerations (Figure 9). This report is only an in vitro study and representative case presentation. The free-radical-scavenging activity of sedative agents in vivo studies and in critically ill patients has not yet been studied. Further studies, including in vivo studies and clinical trials involving the effects on oxidative stress and oxidation and antioxidant balance, are needed.

## Figures and Tables

**Figure 1 biomedicines-11-02129-f001:**
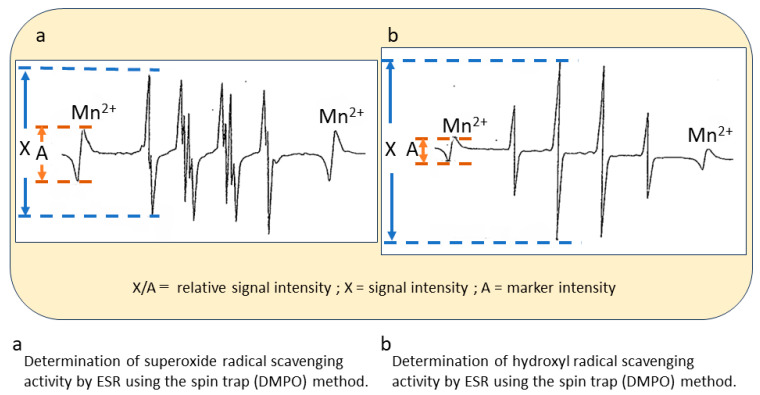
**ESR signals of hydroxyl radicals and superoxide radicals**. Determination of hydroxyl radical (**a**) and superoxide radical (**b**) scavenging activities with electron spin resonance using the spin trap method. MnO was used as an external standard. X, signal intensity; A, marker intensity; X/A = 100, relative signal intensity.

**Figure 2 biomedicines-11-02129-f002:**
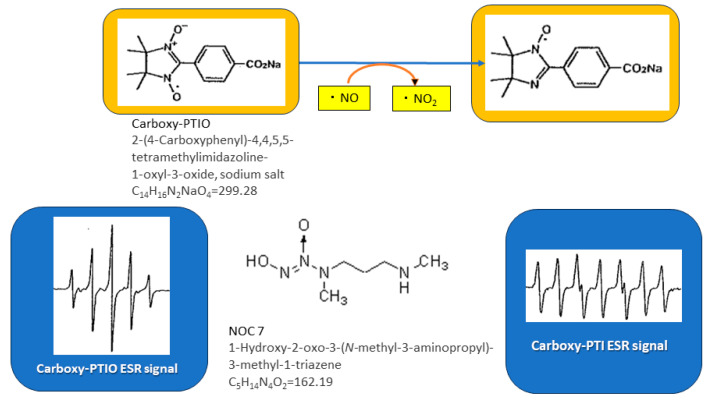
Scheme describing NO reaction using ESR method. Nitric oxide-scavenging activity measured using electron spin resonance spectroscopy. The penta-signal of carboxy-PTIO changed to the hepta-signal of carboxy-PTI.

**Figure 3 biomedicines-11-02129-f003:**
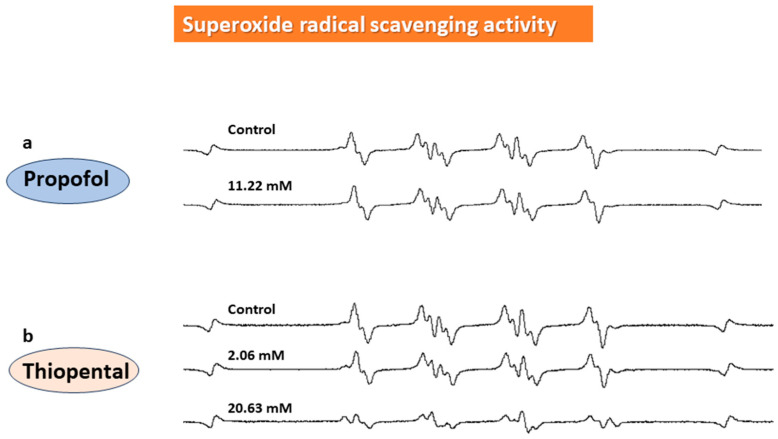
**Superoxide-radical-scavenging activity of sedatives**. As shown using ESR, the formation of the superoxide radical–DMSO spin adduct was strongly inhibited with thiopental (**b**). Propofol (**a**) did not have direct superoxide-radical-scavenging activity.

**Figure 4 biomedicines-11-02129-f004:**
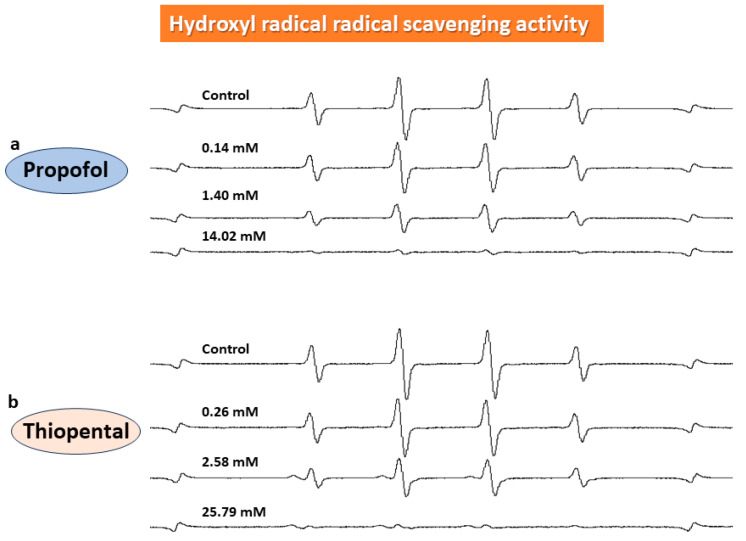
**Hydroxy-radical-scavenging activity.** As shown using ESR, the formation of the hydroxyl radical–DMSO spin adduct was strongly inhibited with propofol (**a**) and thiopental (**b**).

**Figure 5 biomedicines-11-02129-f005:**
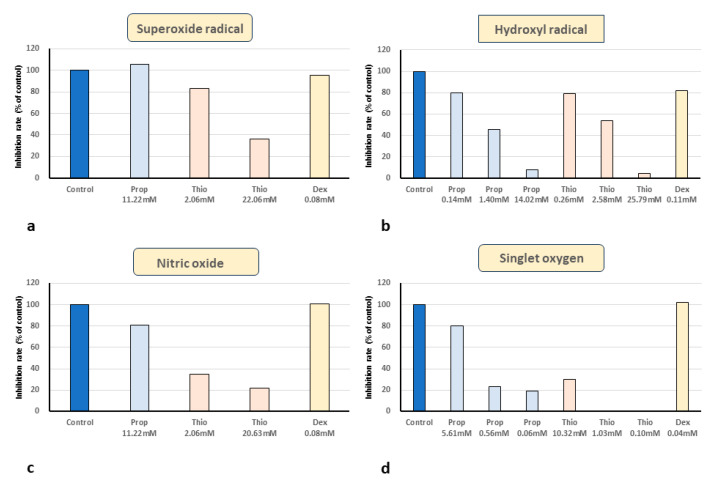
**Direct free-radical-scavenging activities of propofol, thiopental, and dexmedetomidine**. (**a**) Thiopental has direct superoxide-radical-scavenging activity. (**b**) Hydroxyl-radical-scavenging activity of sedatives. Dexmedetomidine, propofol, and thiopental scavenged hydroxyl radicals. (**c**) Nitric oxide-scavenging activity of sedatives: NO was scavenged by propofol and thiopental. (**d**) Singlet oxygen-scavenging activity of sedatives: propofol and thiopental scavenged singlet oxygen.

**Figure 6 biomedicines-11-02129-f006:**
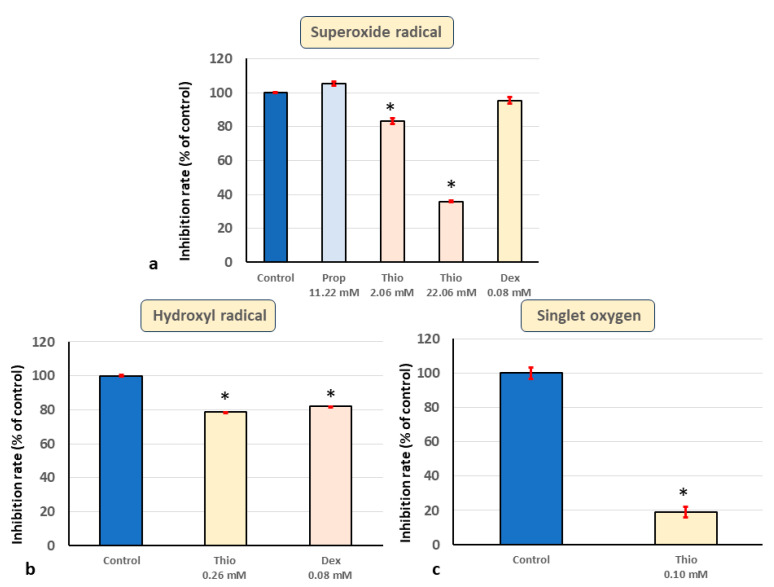
**Direct free-radical-scavenging activities of DEX, thiopental, and dexmedetomidine**. (**a**) Thiopental significantly scavenged superoxide radicals. * *p* < 0.01. (**b**) Hydroxyl-radical-scavenging activity of thiopental and dexmedetomidine. Thiopental and dexmedetomidine significantly scavenged hydroxyl radicals. * *p* < 0.01. (**c**) Singlet oxygen-scavenging activity of thiopental. Thiopental scavenged singlet oxygen significantly. * *p* < 0.01. Error bar (red) is standard error, SE.

**Figure 7 biomedicines-11-02129-f007:**
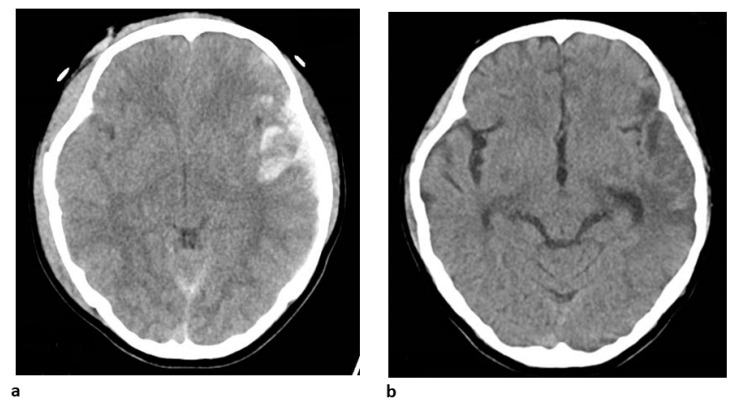
**CT images from a case of severe traumatic brain injury**. CT on admission (**a**) showing diffuse brain swelling, left acute subdural hematoma, and contusion. The ambient cistern is very narrow, and impending cerebral herniation can be observed. Brain swelling was ameliorated, and hematoma decreased in CT after treatment (**b**).

**Figure 8 biomedicines-11-02129-f008:**
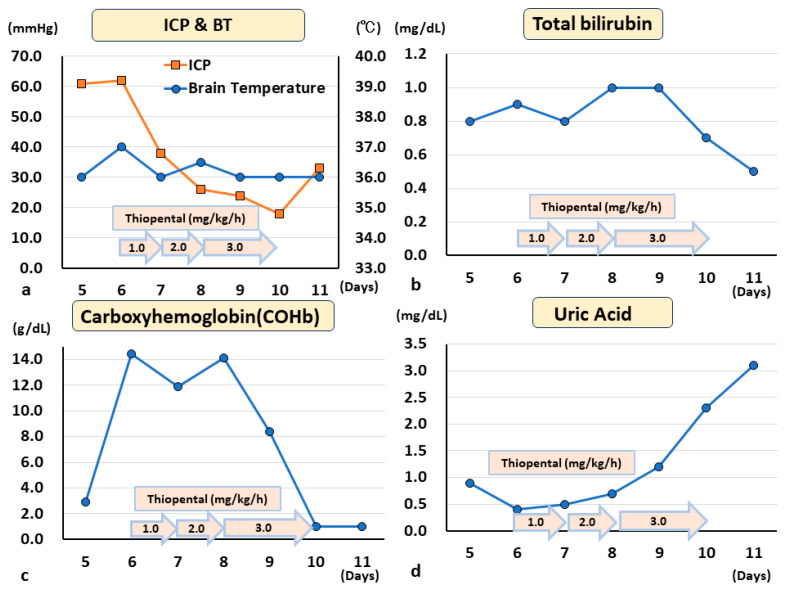
**Endogenous oxidant and antioxidant biomarkers in a patient with traumatic brain injury receiving thiopental treatment**. Thiopental administration markedly reduced ICP (**a**). The blood bilirubin (**b**) and COHb (**c**) concentrations were similar. Both were decreased with thiopental administration. Thiopental may suppress the stress protein HO-1. The blood concentration of uric acid, an endogenous antioxidant (**d**), markedly increased with thiopental administration. Thus, the rapid changes in uric acid may have been affected by thiopental administration.

**Figure 9 biomedicines-11-02129-f009:**
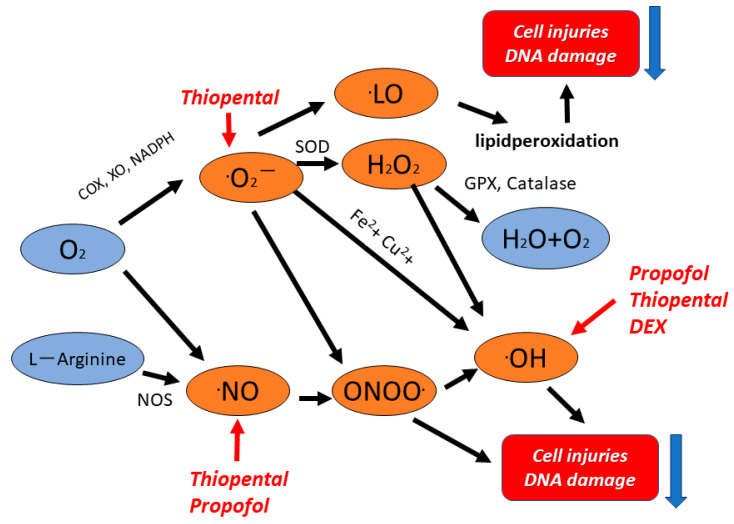
**Scheme showing free radical generation under various critical conditions and the role of sedatives**. Superoxide radicals contribute to the production of other reactive oxygen species (ROS). Thiopental directly scavenges superoxide radicals, the source of other ROS. Therefore, thiopental may indirectly decrease the production of other ROS.

## Data Availability

Not applicable.

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
