# Peer review of "Sedation Therapy in Intensive Care Units: Harnessing the Power of Antioxidants to Combat Oxidative Stress"

_biomedicines, 2023, doi:10.3390/biomedicines11082129_

Round 1

Reviewer 1 Report

Thank you for permitting me to review this manuscript

In this manuscript , the authors assessed the scanvenging properties of some pharmacological sedative agent with regard to antioxidant capacity 

Although the antioxydant therapy in some sedatives are reported but none of these are used for this purpose and all of them are used for sedation only , therefor authors shoul elaborate some practical hypothesis for these paper 

Line 49  there is a repetition 

clinical case page 6 , how are we sure that  this would not happen  naturally if  propofol was not used , is there any animal study describing similar results ?, 

Figure 7 a better pointing of arrows with exact site of action 

The authors should give some axis for future clinical  research as their own statement highlighted that thse are only in vitro studies 

Author Response

Reviewer 1

Reply to Comments

I appreciated reviewing our manuscript.

  1. clinical case page 6 , how are we sure that  this would not happen  naturally if  propofol was not used , is there any animal study describing similar results ?, 

Thank you for pointing this out. We agree with this comment. In clinical practice, the prevailing condition of the patient and multiple other factors can affect a variety of data. Therefore, although it is possible that the changes in UA, CO, and bilirubin data may not be entirely due to thiopental, at the time of data sampling for this patient, no other oxidative stress-related treatments were being administered and there were no pathological changes that would have caused such a drastic change. There were no other treatments related to oxidative stress at the time of data sampling. Therefore, we have added the below provided text as a limitation in the revised manuscript.

This study was only a preliminary case presentation. In clinical practice, data may be influenced by a variety of factors and the condition of the patient prevailing at the time. Therefore, although the changes in UA, CO, and bilirubin data may not be solely due to thiopental administration, no other oxidative stress-related treatment was being administered to this patient at the time the data were collected. Moreover, there were no pathological changes that would cause such a drastic change.”

  1. Figure 7 a better pointing of arrows with exact site of action

Thank you very much for pointing that out. We have revised this to make the figure clear to interpret.

  1. The authors should give some axis for future clinical  research as their own statement highlighted that thse are only in vitro studies 

Thank you for pointing this out. We agree with this comment. We have re-structured the entire document and added measures and issues related with future clinical research.

Reviewer 2 Report

Inoue and colleagues used electron spin resonance to evaluate the antioxidant effects of known sedatives. A hallmark of the study is the evaluation of free radical scavenging activity of sedatives against various ROS (Superoxide, hydroxyl radical and singlet oxygen) and RNS (represented by NO). In general, the authors obtained interesting results. However, I have questions about the study design and discussion.

Comments:

1. Authors write in a number of places only about ROS. However, NO is a representative of the RNS. The terminology needs to be corrected.

2. Figure 5. It is not clear what values are shown on the y-axis. In addition, this figure suggests that the experiments were done in one replicate. It is necessary to supplement them and conduct an adequate statistical analysis.

3. The authors do not discuss the mechanisms underlying the antioxidant action of the studied sedatives. It is necessary to discuss why the studied agents have scavenging activity only against certain free radicals and do not affect others.

Author Response

Reviewer 2

Reply to Comments

I appreciated reviewing our manuscript.

  1. Authors write in a number of places only about ROS. However, NO is a representative of the RNS. The terminology needs to be corrected.

Thank you for pointing this out. We agree with that NO· is a free radical, but it is RNS and not ROS. We have revised the terminology and added the following documents and references regarding RNS:

Reactive nitrogen species (RNS) is a general term for highly reactive nitrogen oxides, such as nitric oxide (NO) and peroxynitrite (ONOO). Similar to reactive oxygen species, they cause oxidative damage to DNA, proteins, and membranes, showing strong cytotoxicity. D. A. Wink & J. B. Mitchell: Free Radic. Biol. Med., 25, 4 (1998).

  1. Figure 5. It is not clear what values are shown on the y-axis. In addition, this figure suggests that the experiments were done in one replicate. It is necessary to supplement them and conduct an adequate statistical analysis.

Thank you for pointing this out. In this graph, the Y-axis represents the relative signal intensity (RSI). We have added this to the graph. Initially, we assumed that data analysis of multiple experiments was necessary; however, later we came to the conclusion that there is no inherent problem with a single experiment because in in vitro ESR experiments, similar results can be obtained even with repeated experiments, and changes in the radical spin adduct diagram are more important than actual the numerical values. However, as you stated, a statistical analysis would be more scientific. For the key data in this case, we have repeated the examination with three replicates and included statistical analysis (Fig. 6). For the other results, only single experiments are presented because would have missed the 10-day re-submission deadline due to all reexaminations.

  1. The authors do not discuss the mechanisms underlying the antioxidant action of the studied sedatives. It is necessary to discuss why the studied agents have scavenging activity only against certain free radicals and do not affect others.

Thank you for pointing this out. We strongly agree with your comment. The explanation is simple for drugs, in terms of structural formula, with reducing ability. However, the issue with in vitro ESR is that it can only identify the presence or absence of direct scavenging ability. Sometimes, there are cases in which the chemical structure shows the potential for scavenging, but scavenging does not occur, or vice versa. Therefore, it is necessary to conduct fundamental studies to clarify whether each radical has a different scavenging ability in future. However, we were unable to include this information in this study. We understand that this is an important omission and a limitation of this study; hence, we have included a relevant statement to the Discussion section in the revised manuscript.

It is also very important to clarify the pharmacological mechanism of action of each drug on antioxidant activity. However, the in vitro ESR method employed in this study can only measure the direct scavenging activity of each reactive oxygen species and NO. The mechanism of the scavenging activity of each drug against each free radical observed in this study requires further investigation.”

Round 2

Reviewer 2 Report

The authors generally responded to my comments. The work may be accepted.

Author Response

I appreciated all of your comments and efforts.

Thanks,

Kenji Dohi MD., PhD.